# The association between fetal fraction and pregnancy-related complications among Chinese population

**Yan Jiang[1☯], Yidan Zhang[2☯], Qin Yang[3], Dan Zeng[4], Keyan Zhao[2], Xin Ma[2,5]\*, Wei Yin[1]\***

**1** Department of Obstetrics, Mianyang People's Hospital, Mianyang, Sichuan, China, **2** GenomCan Inc., Chengdu, Sichuan, China, **3** Department of Clinical Laboratory, Mianyang People's Hospital, Mianyang, Sichuan, China, **4** Chengdu CapitalBio Technology Co., Ltd., Chengdu, Sichuan, China, **5** Department of Statistics, Stanford University, Stanford, CA, United States of America

☯ These authors contributed equally to this work.
\* xin.ma@genomcan.com (XM); wei_yin1@163.com (WY)

**Data Availability Statement:** All relevant data are within the paper and its Supporting Information files.

## Abstract

To examine the association of fetal fraction with a wide spectrum of pregnancy-related complications among Chinese population, we carried out a single-institution retrospective cohort study of women with negative Noninvasive prenatal testing (NIPT) results and singleton pregnancies between May 2018 and May 2020. Indicators of pregnancy-related complications were examined individually, including preterm birth, low birth weight, hypertensive disorders of pregnancy, gestational diabetes, oligohydramnios and intrahepatic cholestasis. We evaluated disease odds ratios (ORs) and 95% confidence intervals (CIs), after controlling for potential confounders including body mass index (BMI), maternal age and gestational week at NIPT. A total of 3534 women were included in our analyses. Women with fetal fraction<15.15% had increased risk of gestational hypertension (OR 4.41, CI [1.65, 12.45]) and oligohydramnios (OR 2.26, CI [1.33, 3.80]) compared to women with fetal fraction≥15.15%. No significant associations with fetal fraction were found for preterm birth, low birth weight, gestational diabetes, and intrahepatic cholestasis. In Summary, fetal fraction is inversely associated with the risk of gestational hypertension and oligohydramnios.

## Introduction

Pregnancy-related complications are the leading cause of the morbidity and mortality of pregnancy. Nearly one fifth of all pregnant women are hospitalized before delivery due to complications [1]. Common pregnancy-related complications include hypertensive disorders of pregnancy (HDP) and gestational diabetes [2–4]. In addition, other complications such as intrahepatic cholestasis are associated with some specific adverse pregnancy outcomes including preterm birth and pregnancy loss [5]. It is desirable to identify patients at risk for pregnancy-related complications before the onset of symptoms to better manage diseases and reduce mortality.

Fetal cell-free DNA (cfDNA) is a candidate biomarker for pregnancy related complications. Cell free DNA in the blood plasma of pregnant women is composed of DNA fragments mainly

**Funding:** XM was supported by grant 2019-YF09-00034-CG funded by Chengdu Science and Technology Bureau (http://cdst.chengdu.gov.cn/). The funders had no role in study design, data collection and analysis, decision to publish, or preparation of the manuscript.

**Competing interests:** YZ, KZ, and XM are employees of GenomCan Inc.. DZ is an employee of Chengdu CapitalBio Technology Co., Ltd.. The remaining authors do not have any conflicts of interest to disclose. This does not alter our adherence to PLOS ONE policies on sharing data and materials.

from the mother herself, but also from the fetus. Fetal cfDNA contributes approximately 10–20% of the total cfDNA [6]. Fetal cfDNA is released into the maternal circulation via apoptosis of placental villous trophoblasts throughout gestation [7]. Noninvasive prenatal testing (NIPT) has become a standard screening test for trisomy 21, 18, 13 and other selected chromosomal abnormalities since its introduction into the clinic in the early 21st century [8]. With the development and commercial availability of NIPT, fetal and maternal cfDNA can be sequenced efficiently and cost-effectively through massively parallel DNA sequencing (MPS). Other than its utility in testing the chromosomal abnormalities of the fetus, NIPT sequencing data can also give reliable fetal fraction estimation in the maternal plasma, defined as the proportion of fetal cfDNA among the total amount of cfDNA, which usually exceeds 4% beginning at 10 weeks' gestation and peaks between 10 and 21 weeks of gestation [7].

Previous studies have investigated the relationship between fetal fraction and various pregnancy complications including HDP [9–13], gestational diabetes [14], and preterm birth [10–12, 15]. However, the significance and directionality of these associations were inconsistent. In addition, most of previous studies were conducted in European and American populations. Studies in East Asians, especially in Chinese populations are limited. Thus, combing the rich clinical information and sequencing data from a large single-center NIPT study, we can assess the relationship between fetal fraction and the subsequent development of pregnancy-related complications in Chinese women.

## Materials and methods

### Study design and data collection

We performed a single-institution retrospective cohort study of all women that underwent NIPT screening from May 2018 to May 2020 at Mianyang People's Hospital. Women were recruited without any prior indication of genetic diseases or fetal structural abnormalities, following the guideline by the National Health and Family Planning Commission of China (http://www.nhc.gov.cn/ewebeditor/uploadfile/2016/11/20161111103703265.docx). Genetic counseling was provided before the NIPT test and when returning the NIPT result. Only women who had singleton pregnancies and showed no chromosomal aneuploidies from NIPT and with valid fetal fraction estimation and pregnancy complications information were included. Further exclusion criteria were: (1) women with multiple pregnancies; (2) women with missing information on whether singleton or multiple pregnancy; (3) women with missing fetal fraction information; (4) women with a history of chronic hypertension or diabetes.

All NIPT sequencing was performed by the same laboratory (Mianyang People's Hospital). The cell-free DNA extraction, library construction, sequencing and bioinformatics analysis were performed using protocols as described in a previous study [16]. In brief, the Bioelectron-Seq 4000 sequencing instrument and the Sequencing Reaction Universal Kit (CapitalBio, Dongguan, People's Republic of China) were used for MPS of plasma cfDNA fragments. Sequencing reads were filtered and aligned to the human reference genome (hg19). The fetal fraction was estimated based on the different fragment length distributions between maternal and fetal cfDNA. Fetal cfDNA fragment tend to have a higher proportion of short plasma DNA fragments (∼130–140 bp; region A) and a lower proportion of long plasma DNA fragments (∼155–175 bp; region B). Locally weighted scatterplot smoothing (LOESS) regression was applied to fit the fetal fraction against reads ratio in features A and B. Male fetuses with fetal fraction estimated from Y chromosome reads were used to train the model parameters [17].

The demographic characteristics and clinical information on pregnancy-related complications were extracted from electronic medical records. Body mass index (BMI) was calculated

from the height and weight data recorded at the time of NIPT test. Parity was dichotomized to either nulliparous or multiparous. Written informed consent was obtained from all participating women and the study received approval from the Institutional Review Board at Mianyang People's Hospital (No. 20201008).

The outcomes of interest were pregnancy-related complications, including low birth weight (<2500g), preterm birth (<37 weeks), gestational diabetes, hypertensive disorders of pregnancy (HDP), oligohydramnios, and intrahepatic cholestasis. HDP include gestational hypertension and preeclampsia. Gestational hypertension was defined as having a systolic blood pressure greater than 140 mmHg or diastolic blood pressure greater than 90 mmHg after 20 weeks of gestation without the evidence of proteinuria, and the blood pressure returning to normal within 12 weeks after the delivery. Preeclampsia was defined as meeting the aforementioned blood pressure criteria plus any of the following complications after the 20th week of pregnancy: proteinuria greater than or equal to 300 mg per 24-hour urine collection; a low platelet count; impaired liver function; signs of kidney problems indicated by the serum creatinine level; pulmonary edema and new-onset cerebral or visual disturbances. Gestational diabetes was defined as diabetes firstly seen in a pregnant woman who had no diabetes before the pregnancy, and was diagnosed by an initial glucose challenge test followed by a 75g oral glucose tolerance test. Intrahepatic cholestasis of pregnancy was defined as severe itchiness and increased serum bile acid concentrations. Oligohydramnios was defined as a single deepest vertical pocket (DVP) of less than 2cm on ultrasound examination at 28–40 weeks.

We analyzed the variable of interest, fetal fraction, as both continuous and dichotomous variables. Since the directionality of the associations between fetal fraction and pregnancy complications were conflicting from previous studies, we firstly treated fetal fraction as a continuous variable to determine the significance and directionality of the relationships. We then carried out analyses treating fetal fraction as a dichotomous variable. A threshold of low or high fetal fraction was defined based on its distribution within the cohort, the results from the associations between continuous fetal fraction and the outcomes of interest, as well as existing literatures [11].

## Statistical analysis

Baseline characteristics were compared between the two groups of dichotomous fetal fraction. We compared categorical data using chi-square test and continuous data using student t-test. We calculated odds ratios (ORs) and 95% confidence intervals (CIs) using multivariable logistic regression to determine the associations between fetal fraction and pregnancy-related complications, controlling for gestational age at NIPT, maternal age and BMI. Each pregnancy complication was examined individually. We firstly assessed the association between fetal fraction with HDP, combining gestational hypertension and preeclampsia. We also evaluated the relationships of fetal fraction with gestational hypertension and preeclampsia separately, to show which specific disease contributes more to the fetal fraction-HDP association. For all association tests with each complication, only normal samples were included in the control group.

We hypothesized that relationships of fetal fraction with pregnancy complications could be different for women undergoing NIPT in different trimesters of pregnancy. To examine this hypothesis, we performed stratified analyses based on gestational age at NIPT: first trimester (10–14.9 weeks), second trimester (15 and 22 weeks) and third trimester (>22 weeks) in order to find the optimal time to perform the NIPT screening tests. All analyses were conducted using R version 4.0.3 (R Core Team, 2020). A P<0.05 from the association test was considered statistically significant.

## Results

### Basic characteristics of study participants

There are 3834 women who underwent NIPT at Mianyang People's Hospital between May 2018 and May 2020. Twenty-six women had NIPT chromosomal abnormality (a positive rate of 0.68%), which were excluded from the study. We also excluded 8 women with a history of chronic hypertension or diabetes. We finally obtained 3534 cases in the study after further excluding 266 patients without NIPT results, with multiple pregnancy and with missing information on whether singleton or multiple pregnancy. Details regarding the excluded cases were shown in Fig 1.

Fetal fractions within the study population ranged from 4.32% to 50.38%, with a mean value of 18.48% and a standard deviation of 5.28%. The dichotomization cut-off point for fetal fraction was set to the 25th percentile, namely 15.15%. The choice of the cut-off point is based on the results from the distributions of fetal fraction stratified by the traits showing significant

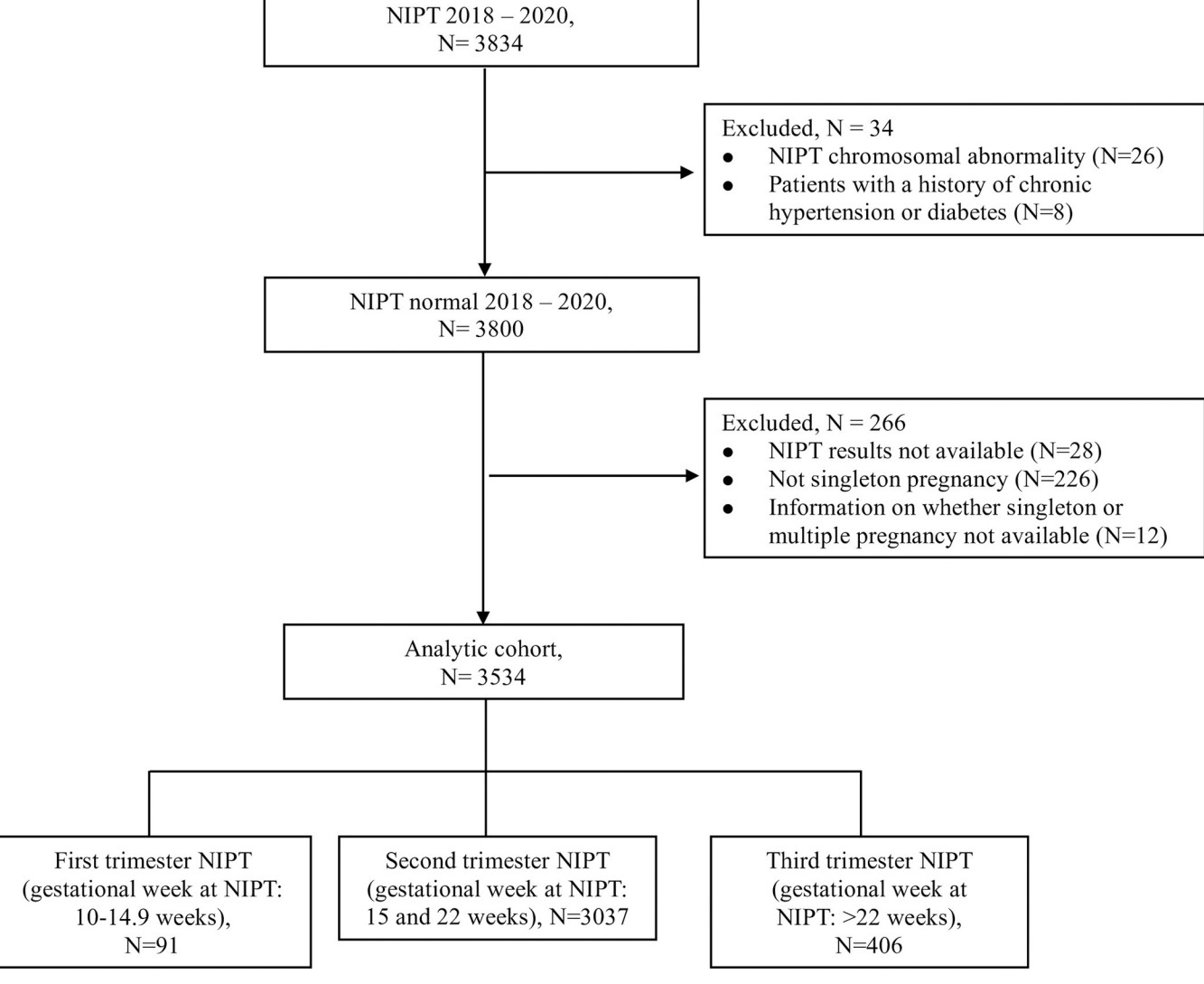

**Fig 1. Flow chart defining the study population.** The flow chart illustrates the filtering steps applied in the study. NIPT, noninvasive prenatal testing.

**Table 1. Characteristics of pregnant women with low fetal fraction compared with women with normal fetal fraction.**

| | Fetal fraction≥25th percentile (N = 2650) | Fetal fraction<25th percentile (N = 884) | P value |
|---|---|---|---|
| **Age** (mean (SD)) | 28.51 (3.94) | 28.67 (4.15) | 0.297 |
| **Gestational age at NIPT** (mean (SD)) | 17.77 (2.79) | 17.64 (2.61) | 0.217 |
| **Parity**, n (%) | | | |
| multiparous | 984 (37.1%) | 335 (37.9%) | 0.714 |
| nulliparous | 1666 (62.9%) | 549 (62.1%) | |
| **IVF-ET pregnancy**, n (%) | | | |
| no | 2604 (98.3%) | 868 (98.5%) | 0.765 |
| yes | 45 (1.7%) | 13 (1.5%) | |
| **BMI** (mean (SD)) | 21.89 (2.90) | 23.30 (3.52) | **<0.001** |
| **Gestational age at delivery**, (mean (SD)) | 39.44 (1.03) | 39.34 (1.13) | **0.010** |
| **Method of delivery**, n (%) | | | |
| cesarean | 1313 (49.5%) | 476 (53.8%) | **0.030** |
| normal | 1337 (50.5%) | 408 (46.2%) | |

NIPT, noninvasive prenatal testing; IVF-ET, In Vitro Fertilization and Embryo Transfer; BMI, body mass index

associations with continuous fetal fraction, which is mentioned below, as well as previous literatures [11]. The 25th percentile cut-off is also widely used in stratifications in statistical analysis. Baseline characteristics of the two groups of dichotomous fetal fraction are shown in Table 1. We found statistically significant differences between the two groups regarding maternal BMI, gestational age at delivery, and method of delivery. This is not surprising given that associations of fetal fraction with gestational age and maternal BMI are well established in previous publications [18, 19]. However, the distributions of maternal age and gestational age at NIPT, and the percentages of nulliparous and IVF-ET pregnancies were similar between the two groups of pregnant women with the dichotomous fetal fraction.

## The association of fetal fraction with pregnancy-related complications

Table 2 shows associations between continuous fetal fractions and the pregnancy outcomes. The increase of fetal fraction had significant protective effects on hypertensive disorder of pregnancy (HDP), intrahepatic cholestasis and oligohydramnios. When dividing HDP into gestational hypertension and preeclampsia, a significant association was only observed for gestational hypertension. After adjusting for BMI, maternal age and gestational age at NIPT, only

**Table 2. Relationships between fetal fraction (continuous variable) and pregnancy-related complications.**

| Disease (N) | Crude OR (95% CI) | P value | Adjusted OR [a] (95% CI) | P value |
|---|---|---|---|---|
| Low birth weight (N = 78) | 1.00 (0.96, 1.04) | 0.883 | 1.01 (0.95, 1.07) | 0.788 |
| Preterm birth (N = 64) | 1.01 (0.96, 1.05) | 0.706 | 1.05 (0.98, 1.11) | 0.169 |
| HDP (N = 71) | 0.93 (0.89, 0.98) | **0.00764** | 0.92 (0.85, 1.00) | 0.0524 |
| Gestational Hypertension (N = 29) | 0.89 (0.81, 0.96) | **0.00591** | 0.89 (0.78, 0.99) | **0.0484** |
| Preeclampsia (N = 42) | 0.96 (0.90, 1.02) | 0.21 | 0.95 (0.85, 1.05) | 0.380 |
| Gestational diabetes (N = 459) | 0.98 (0.96, 1.00) | 0.0857 | 1.00 (0.98, 1.03) | 0.814 |
| Intrahepatic cholestasis (N = 148) | 0.96 (0.93, 0.99) | **0.0149** | 0.97 (0.93, 1.02) | 0.292 |
| Oligohydramnios (N = 113) | 0.95 (0.92, 0.99) | **0.0205** | 0.96 (0.90, 1.01) | 0.0922 |

a. Adjusted for body mass index (BMI), gestational age at NIPT and maternal age

NIPT, noninvasive prenatal testing; HDP, Hypertensive disorder of pregnancy; OR, odds ratio; CI, confidence interval

**Table 3. Relationship between fetal fraction (Fetal fraction<25th percentile vs. fetal fraction≥25th percentile [a]) and pregnancy-related complications.**

| Disease (N) | Crude OR (95% CI) | P value | Adjusted OR [b] (95% CI) | P value |
|---|---|---|---|---|
| Low birth weight (N = 78) | 1.11 (0.65, 1.81) | 0.695 | 0.89 (0.35, 1.93) | 0.775 |
| Preterm birth (N = 64) | 1.48 (0.85, 2.47) | 0.148 | 1.30 (0.52, 2.94) | 0.546 |
| HDP (N = 71) | 2.65 (1.64, 4.27) | **<0.001** | 2.49 (1.22, 5.02) | **0.0111** |
| Gestational Hypertension (N = 29) | 3.76 (1.80, 8.01) | **<0.001** | 4.41 (1.65, 12.45) | **0.00343** |
| Preeclampsia (N = 42) | 2.08 (1.09, 3.85) | **0.0218** | 1.45 (0.50, 3.86) | 0.467 |
| Gestational diabetes (N = 459) | 1.12 (0.89, 1.40) | 0.341 | 0.82 (0.56, 1.19) | 0.310 |
| Intrahepatic cholestasis (N = 148) | 1.33 (0.92, 1.91) | 0.121 | 1.25 (0.71, 2.10) | 0.419 |
| Oligohydramnios (N = 113) | 1.95 (1.31, 2.87) | **0.000845** | 2.26 (1.33, 3.80) | **0.0022** |

a. Fetal fraction≥25th percentile as reference group

b. Adjusted for body mass index (BMI), gestational age at NIPT and maternal age

HDP, Hypertensive disorder of pregnancy; OR, odds ratio; CI, confidence interval

the association with gestational hypertension remained significant (P = 0.0484) (Table 2). For every one percent increase of fetal fraction, the odds of gestational hypertension decreased by a factor of 0.89 (OR = 0.89, 95% CI = [0.78, 0.99]). We found no association between the continuous fetal fraction and risks of low birth weight, preterm birth, and gestational diabetes. The fetal fraction distributions comparing women with gestational hypertension versus normal women, women with oligohydramnios versus normal women (S1 and S2 Figs) reveal that the maximum segregation of distribution occurs around the 25[th] percentile (fetal fraction = 15.15%).

The associations of the dichotomous fetal fractions (<25[th] percentile and ≥25[th] percentile) with pregnancy outcomes are presented in Table 3. We found significant associations of fetal fraction with the risk of HDP and oligohydramnios. For HDP, when considering gestational hypertension and preeclampsia as separate diseases, both were significantly associated with fetal fraction. When adjusted for BMI, maternal age and gestational age at NIPT, the associations with HDP and oligohydramnios remained significant. Compared to pregnant women with fetal fraction≥15.15%, women with fetal fraction<15.15% were 2.49 times more likely to have HDP (OR = 2.49, 95% CI = [1.22, 5.02], P = 0.0111) during pregnancy (Table 3). However, similar to the analyses using continuous fetal fraction, only the association with gestational hypertension remained significant after controlling for potential confounders (P = 0.00343), and the association with preeclampsia became insignificant (P = 0.467). For women with fetal fraction<15.15%, we saw 3.41 times increase in the odds of getting gestational hypertension, compared to women with fetal fraction≥15.15% (OR = 4.41, 95% CI = [1.65, 12.45]). The odds ratio of oligohydramnios for pregnancy women in fetal fraction<15.15% group compared with the fetal fraction≥15.15% group was 2.26 (95% CI = [1.33, 3.80], P = 0.0022), after adjusting for BMI, maternal age and gestational age at NIPT (Table 3). For other traits, we found that fetal fraction was not statistically significantly associated with low birth weight, preterm birth, gestational diabetes and intrahepatic cholestasis.

## Subgroup analyses

Previous studies showed that the patterns of associations were different when women performed NIPT tests at different trimesters [9, 15]. Thus, we conducted stratified analyses of the associations between fetal fraction and pregnancy complications according to gestational age at NIPT. Ninety-one pregnant women underwent NIPT during the first trimester, three thousand thirty-seven women during the second trimester and four hundred six women during the

third trimester (Fig 1). For women who underwent NIPT during the first trimester (<15 weeks), only the trait of gestational diabetes had enough samples (n = 13) for the data analysis. All outcomes of interest had enough observations among the cohort including women who underwent NIPT during the second trimester (15 and 22 weeks). For women who underwent NIPT during the third trimester (>22 weeks), only 2 traits: gestational diabetes (n = 50) and intrahepatic cholestasis (n = 19) had enough samples for the data analysis.

After adjusting for BMI and maternal age, women in the fetal fraction<25th percentile group during the second trimester were significantly more likely to develop HDP and oligohydramnios than women in the fetal fraction≥25th percentile group (Table 4). Women with fetal fraction <15.15% had more than 2 times the odds of having HDP compared to women whose fetal fraction with ≥15.15% (OR = 2.19, 95%CI = 1.03–4.58, P = 0.0386). Specifically, the odds ratio of gestational hypertension for pregnant women in fetal fraction<15.15% group compared with fetal fraction≥15.15% group was 3.32 (95%CI = 1.16–9.76, P = 0.0242, Table 4). However, fetal fraction at the second trimester showed no significant association with preeclampsia after controlling for potential confounders (Table 4). At second trimester, women with fetal fraction<15.15% showed 1.13 times increase in the odds of getting oligohydramnios compared to women with fetal fraction≥15.15% after controlling for BMI and maternal age (OR = 2.13, 95%CI = 1.20–3.70, P = 0.008, Table 4). For women undergoing NIPT during the first and the third trimester, no significant associations were found between fetal fraction and pregnancy-related complications (Table 4).

**Table 4. Relationship between fetal fraction (Fetal fraction<25th percentile vs. fetal fraction≥25th percentile [a]) and pregnancy-related complications stratified by gestational age at NIPT.**

| Disease (N) | Crude OR (95% CI) | P value | Adjusted OR [b] (95% CI) | P value |
|---|---|---|---|---|
| **First Trimester** | | | | |
| Gestational diabetes (N = 13) | 1.44 (0.34, 5.35) | 0.592 | 0.61 (0.0065, 25.43) | 0.799 |
| **Second Trimester** | | | | |
| Low birth weight (N = 65) | 1.13 (0.64, 1.93) | 0.658 | 0.67 (0.22, 1.62) | 0.412 |
| Preterm birth (N = 52) | 1.87 (1.04, 3.26) | **0.0304** | 1.78 (0.69, 4.24) | 0.210 |
| HDP (N = 63) | 2.69 (1.62, 4.47) | **0.000127** | 2.19 (1.03, 4.58) | **0.0386** |
| Gestational Hypertension (N = 26) | 3.45 (1.58, 7.65) | **0.00181** | 3.32 (1.16, 9.76) | **0.0242** |
| Preeclampsia (N = 37) | 2.26 (1.15, 4.35) | **0.0156** | 1.56 (0.53, 4.25) | 0.393 |
| Gestational diabetes (N = 396) | 1.10 (0.85, 1.40) | 0.466 | 0.86 (0.57, 1.27) | 0.447 |
| Intrahepatic cholestasis (N = 126) | 1.33 (0.89, 1.95) | 0.158 | 1.21 (0.67, 2.10) | 0.508 |
| Oligohydramnios (N = 101) | 1.57 (1.02, 2.38) | **0.0371** | 2.13 (1.20, 3.70) | **0.008** |
| **Third Trimester** | | | | |
| Gestational diabetes (N = 50) | 1.22 (0.57, 2.47) | 0.586 | 0.40 (0.06, 1.50) | 0.235 |
| Intrahepatic cholestasis (N = 19) | 1.38 (0.43, 3.82) | 0.553 | 0.79 (0.040, 5.42) | 0.839 |

a. Fetal fraction≥25th percentile as reference group

b. Adjusted for body mass index (BMI) and maternal age

NIPT, noninvasive prenatal testing; HDP, Hypertensive disorder of pregnancy; OR, odds ratio; CI, confidence interval

## Discussion

In this retrospective cohort study, we observed significant inverse associations of fetal fraction with the risk of hypertensive disorders of pregnancy (HDP) and oligohydramnios after adjusting for potential confounders. The odds of having HDP for women with fetal fraction<15.15% was nearly 3 folds compared to women with fetal fraction≥15.15%. When considering gestational hypertension alone, the odds ratio increased to a value greater than 4. However, when considering eclampsia alone, the association was no more significant. Also, for pregnant women with fetal fraction<15.15%, they were 2.26 times more likely to have oligohydramnios, compared to pregnant women with fetal fraction≥15.15%. We found no associations of fetal fraction with the subsequent development of low birth weight, preterm birth, gestational diabetes and intrahepatic cholestasis.

We observed a significant association of fetal fraction with the subsequent development of oligohydramnios, which contradicted early data from a smaller study [11]. Limited sample size of the previous study may explain the difference, since their study only included 25 oligohydramnios cases in total [11]. Although the exact cause of oligohydramnios is complicated and still unclear, we speculated that subtle placental structural abnormalities could potentially cause oligohydramnios. This could also potentially reduce the release of fetal cfDNA into mother's blood, resulting in low fetal fraction of cfDNA. The association between fetal fraction and oligohydramnios has significant potential clinical value. Oligohydramnios increases the risk of neonatal deaths, stillbirths and low birth weight [20]. Because of the association between oligohydramnios and neonatal abnormalities, surveillance by ultrasound is often recommended to women at risk as early as possible, and when needed, pregnant women are admitted to hospitals for frequent screening and to monitor the signs of preterm labor [21]. A low fetal fraction result in the first trimester or early second trimester may help clinicians identify women who warrant frequent ultrasound screening for the amount of amniotic fluid.

Our findings are in accordance with the results from several previous studies that found lower fetal fraction being associated with a higher risk of HDP. For instance, Gerson et al. found statistically significant association of HDP with lower fetal fraction measured in either the first or the second trimester NIPT in their prospectively enrolled cohort of 639 women [11]. Also, Suzumori et al. found fetal cell-free DNA fraction sampled between 10 and 20 weeks' gestation were associated with HDP in a cohort of more than 5000 Japanese pregnant women [13]. However, in the study by Shook at al., no association of HDP with fetal fraction in the first trimester was found [10]. This is probably due to different definition of fetal fraction cut-offs, since they compared high fetal fraction (≥95th percentile) and normal fetal fraction (between the 5th and 95th percentiles). When we tested gestational hypertension and preeclampsia as separate outcomes of interests, our results are inconsistent with previous studies. Gerson at al. and Kim et al. did not find significant relationship between fetal cell-free DNA or fetal fraction in the first or second trimester and the risk of getting gestational hypertension [11, 22]. In terms of preeclampsia, the results from different studies were inconclusive. We did not find any statistically significant association between fetal fraction and preeclampsia in our cohort. The results agree with the findings of Gerson et al., which showed no association between preeclampsia and low fetal fraction measured in the first or second trimester NIPT screening [11]. Prior studies by Thurik et al., Silver et al. and Poon et al. also found no association of preeclampsia with absolute levels of fetal cfDNA in the first trimester and the development of preeclampsia [23–25]. However, the results from studies by Alberry at al., Yu et al. and Muñoz-Hernández et al. indicated that significantly elevated levels of fetal cfDNA were associated with a risk of preeclampsia [26–28]. These studies did not adequately control for potential confounders including BMI and a history of chronic hypertension which have been

shown to independently affect the association. The discrepancy may also result from the heterogeneity of the definition of preeclampsia, and the severe form of preeclampsia is more likely to be associated with fetal cfDNA [11]. We speculate that the pathophysiologic differences in women with preeclampsia and those with gestational hypertension could lead to different association results. Early placental arteriolopathy are more prevalent in women with preeclampsia [29]. The apoptosis process involved could potentially raise the fetal fraction of cfDNA [30]. There are findings indicating the racial differences in the incidence rate of preeclampsia [31, 32], but how this links to the genetic or environmental causes is still an open question. Finding the exact cause of preeclampsia is still a big scientific challenge. Further studies are merited to find the mechanisms behind the associations of fetal fraction with the different categories of HDP.

Previous reports have suggested that absolute values of cfDNA or fetal fraction do not have prognostic value for preterm birth prediction [10, 11, 15, 25], and our study substantiated those findings. However, some other studies found significant association between fetal fraction or fetal cfDNA and the risk of preterm birth. The study from Dugoff et al. reported a significant association between fetal fraction and preterm birth only among pregnant women who took NIPT in the second trimester of gestation and no significant association among those who took NIPT in the first trimester [15]. In addition, Jakobsen et al. found that high levels of fetal fraction ($\geq$95th percentile) was significantly associated with a higher risk of preterm birth [33]. It has been hypothesized that fetal cfDNA can be pro-inflammatory, which elicits the inflammation-parturition cascade and causes preterm birth [34]. However, *in vivo* studies provided conflicting results. Van Boeckel et al. found that inflammation did not change the amount of fetal cfDNA, and fetal cfDNA was not pro-inflammatory. Thus, fetal cfDNA was unlikely to be a cause of inflammation or preterm birth [35]. On the contrary, the study from Gomez-Lopez et al. showed that fetal cfDNA increased before preterm birth that were induced by systemic inflammation, but not before intra-amniotic inflammation-induced preterm birth [36]. Both observational studies and animal models indicated the controversial relationship between fetal fraction and preterm birth.

We observed no associations of fetal fraction with the risk of gestational diabetes and low birth weight. Our findings support the results established from previous studies [12, 14]. Our findings regarding the association between fetal fraction and intrahepatic cholestasis contradicted the early data implicating high total cfDNA in intrahepatic cholestasis cases [5]. One explanation for this difference is that they were using the total cfDNA and did not discriminate the fetal and maternal cfDNA, thus it was not the same as the fetal fraction.

Our study has several strengths. We used a large cohort of more than 3500 women who underwent routine NIPT from a single institution. The sample size in our study was 5–10 times larger than previous studies investigating similar outcomes of interest, which improved the robustness and accuracy of the estimation of associations. In addition, there is significant variation on the definition of fetal fraction and so far, standardization of measuring fetal fraction is still not available. Different screening products using different quantitation methods likely introduce variability of fetal fraction levels [37]. As all NIPT in our study were performed at the same laboratory and used the same screening kit, we could then minimize the possibility for variation due to technical differences. In addition, we excluded patients with a history of chronic hypertension and controlled for potential confounders such as BMI, maternal age and gestational age at NIPT, which made our results more robust. Prior studies have indicated that the associations between fetal fraction and pregnancy complications were different across races [12]. To our knowledge, our study is one of the first investigating the associations between fetal fraction and a wide spectrum of pregnancy-related complications among Chinese population. Existing studies were mainly done in western populations. Thus, this study

can help us gain a more complete understanding of the associations of cfDNA fetal fraction with the pregnancy complications in a diverse ethnic background.

This study has a few limitations. We intended to assess whether the relationships of fetal fraction with pregnancy-related complications were different for women who undertook NIPT in different pregnancy trimesters. While our study had a relatively large sample size, pregnant women enrolled in our study mainly had NIPT in the second trimester. Thus, we did not have sufficient statistical power to test the associations for women undergoing NIPT in the first and the third trimester. Another limitation is that we could not exclude the possibility of other hidden residual confounding, even though we carefully controlled for a series of known confounders including a history of chronic hypertension, BMI, maternal age and gestational age at NIPT.

In summary, the results of this retrospective cohort study show that women with a fetal fraction<15.15% are at increased risk for gestational hypertension and oligohydramnios after controlling for BMI, maternal age and gestational age at NIPT. Other than the purpose of screening for fetal chromosomal abnormalities, NIPT could also provide cfDNA fetal fraction estimation that could potentially guide the screening of women at high risk of certain pregnancy-related complications.

## Supporting information

**S1 Fig. The distribution of fetal fraction grouped by the gestational hypertension status.** Density plot of fetal fraction from gestational hypertension samples and normal samples in 2 different colors. Dotted lines represent the mean values of fetal fraction from the 2 groups. (TIF)

**S2 Fig. The distribution of fetal fraction grouped by the oligohydramnios status.** Density plot of fetal fraction from oligohydramnios samples and normal samples in 2 different colors. Dotted lines represent the mean values of fetal fraction from the 2 groups. (TIF)

**S1 Dataset.**
(CSV)

## Acknowledgments

We thank all patients and their families for participation in this study.

## Author Contributions

**Conceptualization:** Yan Jiang, Yidan Zhang, Qin Yang, Keyan Zhao, Xin Ma, Wei Yin.

**Data curation:** Yidan Zhang.

**Formal analysis:** Yan Jiang, Yidan Zhang, Keyan Zhao.

**Funding acquisition:** Xin Ma.

**Investigation:** Yan Jiang, Qin Yang, Dan Zeng, Wei Yin.

**Methodology:** Yan Jiang, Yidan Zhang, Qin Yang, Keyan Zhao.

**Project administration:** Yan Jiang, Yidan Zhang.

**Supervision:** Qin Yang, Dan Zeng, Keyan Zhao, Xin Ma, Wei Yin.

**Writing – original draft:** Yan Jiang, Yidan Zhang, Keyan Zhao, Xin Ma, Wei Yin.

**Writing – review & editing:** Yan Jiang, Yidan Zhang, Qin Yang, Dan Zeng, Keyan Zhao, Xin Ma, Wei Yin.

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
