## [Decision Letter · Decision Letter 0]

14 Apr 2022

PONE-D-21-33442The association between fetal fraction and pregnancy-related complications among Chinese populationPLOS ONE

Dear Dr. Yin,

Thank you for submitting your manuscript to PLOS ONE. After careful consideration, we feel that it has merit but does not fully meet PLOS ONE’s publication criteria as it currently stands. Therefore, we invite you to submit a revised version of the manuscript that addresses the points raised during the review process.

We look forward to receiving your revised manuscript.

Kind regards,

Badri Padhukasahasram

Academic Editor

PLOS ONE

Journal Requirements:

I have read the journal's policy and the authors of this manuscript have the following competing interests: YZ, KZ, and XM are employees of GenomCan Inc.. DZ is an employee of Chengdu CapitalBio Technology Co., Ltd.. The remaining authors do not have any conflicts of interest to disclose.

Additional Editor Comment:

Dear Dr.Yin,

The reviews of the manuscript "The association between fetal fraction and pregnancy-related complications among Chinese population" are now available. Based on the comments provided by 2 expert reviewers, I conclude that the article requires major revisions. Reviewers note the value of a large study of this size conducted for the first time for Chinese populations. More methodological and analysis details, and improvements to language and grammar will significantly improve this manuscript.

There are also additional queries and details requested by reviewers for choices made in study design such as gestational age.

I look forward to a revised manuscript that addresses all the major and minor comments from the reviewers.

Sincerely,

Dr. Badri Padhukasahasram

Reviewers' comments:

Reviewer's Responses to Questions

**Comments to the Author**

1. Is the manuscript technically sound, and do the data support the conclusions?

Reviewer #1: Yes

Reviewer #2: Partly

2. Has the statistical analysis been performed appropriately and rigorously? 

Reviewer #1: Yes

Reviewer #2: Yes

3. Have the authors made all data underlying the findings in their manuscript fully available?

Reviewer #1: Yes

Reviewer #2: Yes

4. Is the manuscript presented in an intelligible fashion and written in standard English?

Reviewer #1: Yes

Reviewer #2: Yes

5. Review Comments to the Author

Reviewer #1: Manuscript ID PONE-D-21-33442 entitled "The association between fetal fraction and pregnancy-related complications among Chinese population"

Dear Authors,

My comments are followings:

1. Are there any indications for performing NIPT in this study? Do you provide genetic counseling for those who wish to do so?

2. I think the DISEASE in line 116 should be DISORDER.

3. Why did you perform NIPT around 17 weeks of gestation instead of 10-14 weeks?

4. Please indicate in the Materials and methods section at which gestational week the diagnosis of oligohydramnios was made.

5. Could you list the total number of NIPT performed, the positive rate, and the unreportable results rate in the figure?

6. It is speculated that the difference in the pathogenesis of HDP and preeclampsia affects the results of FF. What would you think about placental remodeling in early gestational age?

7. How do you think the FF tends to be lower in pregnant women with oligohydramnios?

8. Do you think there is a background of racial differences or genetic/environmental influences that cause preeclampsia?

9. I would notice a number of errors of English, so that the manuscript should be checked by native speakers of English.

Reviewer #2: The manuscript entitled “The association between fetal fraction and pregnancy-related complications among Chinese population” by Jiang et al. investigates the relationship between fetal fraction and a variety of pregnancy related complications. As stated by the authors, the primary novelty of the work is the fact that few studies of this size have been conducted on the Chinese population, although similar studies have been conducted in European populations. The primary finding of the study was an association between low fetal fraction and increased risk of HDP. An association between low fetal fraction and gestational hypertension was also demonstrated. The odds ratio was larger when stratifying the fetal fraction population by the 25th percentile (15.15%) as opposed to treating fetal fraction as a continuous variable in the logistic regression models.

Although many of the findings in this manuscript have been previously reported, there is value in confirming these findings in a large Chinese population. When a logistic regression model was fit controlling for BMI, gestational age, and maternal age; fetal fraction was only marginally significant (P=0.048) and the confidence interval for the odds ratio nearly covered 1.0. The significance and odds ratio improved dramatically when fetal fraction was dichotomized using the 25th percentile, but no explanation is provided for why this cutoff was chosen. The paper may benefit from more figures showing a comparison of the fetal fraction distributions in the gestation hypertension and normal populations.

The authors also did not provide enough details on some of the methods. The fetal fraction calculation was described very loosely. Either more detail should be provided or a reference to a paper than employed a similar method should be provided. It also wasn’t clear how the control group was defined for the statistical analysis. When evaluating the relationship between gestational hypertension and fetal fraction, were all other diagnoses included? Or only the gestational hypertension samples and the normal samples?

The data was provided, but some more detail on how to analyze it to recreate the data in the paper would be useful. For example, it was not clear how to reproduce the data filtering from the flow chart provided. A list of issues is provided below.

• Figure 1 states there are 3808 total individuals, but the provided data set has 3800 rows.

• Figure 1 states that 226 samples were excluded due to Not singleton pregnancy, however the data has 228 samples with twin == 2.

• Patients with history of hypertension (7) and diabetes (1) are not indicated. Perhaps this explains the difference between the 3808 and 3800?

Finally, there were many grammatical errors in the manuscript. I have tried to address most of them below. Some errors that were made more than once include the unnecessary use of personal pronouns (we, our, etc) and using an incorrect tense.

Line 52: “leading causes” should be “leading cause.” Also, the statement is fairly obvious. “Have been” usually implies some passage of time. Consider: “Pregnancy related complications are the leading cause”.

Lines 53-54: Change “due to the complications” to “due to complications”

Lines 54-56: Consider rephrasing. Start the sentence with: “Common pregnancy related complications include…”

Line 61: cfDNA is a candidate biomarker for pregnancy related complications.

Lines 61-62: Cell free DNA is composed of DNA fragments released into the blood plasma.

Lines 64-66 :Fetal cfDNA is released into the maternal circulation via apoptosis of placental villous trophoblasts throughout gestation.

Line 69: Remove we: “… NIPT, fetal and maternal cfDNA can be sequenced…”

Lines 76: Replace “the links of fetal fraction with” with “the relationship between fetal fraction and”

Line 78: Consider replacing conflicting with inconsistent.

Line 90: Replace countries with populations or perhaps cohorts.

Line 80: Replace “were limited” with “are limited”

Line 88: Replace undergoing with that underwent

Line 89 : Replace receiving with received

Line 99: add “the” before instrument and reagents

Line 101: MPS already defined as an acronym on line 70

Lines 102-105: The method for estimating fetal fraction is not adequately described. Simply writing “estimated using different length distributions” is too vague. Either provide more detail, or provide a reference to a previous method. The reads ratio has not been defined.

Line 103: Add fragment before length

Line 104: Male fetus should be plural

Line 114: Replace “Our” with “The”

Line 126: who did not have..

Line 132: Replace “our” with “the”

Lines 136-137: Please elaborate on how the dichotomous threshold was determined from the FF distribution in the cohort.

Line 148: Unclear what “fetal fraction as a whole” means. If you are not referring to fetal fraction, but rather the combined set of gestational hypertension and preeclampsia, then please clarify.

Lines 148: Replace “Then we went further and” with “We also”

Line 153: replace “our” with “this” and remove “also”

Line 155: Replace “proper” with “optimal.”

Lines 151-155: You discuss stratifying by trimester and identifying the best trimester for testing. Was this done separately for each pregnancy complication? This sentence comes directly after a discussion of hypertensive disease, so it’s unclear if it is just for that complication or for all. If it is for all, consider starting a new paragraph.

Line 162. Generally, numbers greater than 1,000 are written as numbers

Line 162: Replace “our institution” with the institution name.

Line 170: Replace “For our population, the fetal fraction ranged from” with “The fetal fraction within the study population ranged from”

Line 171: Standard deviation needs % units.

Lines 172-175: This sentence is very hard to follow. It has too many commas and it isn’t clear what the subject of the sentence is. If the sentence is meant to describe how the FF cutoff was established, as verb is missing. For example: “The identification of the cut-off point was based”

Lines 176-178: The relationship between FF, gestational age, and BMI is well established. It may be worth mentioning this.

Line 195: add “a”: “For a one percent…”

Line 272: Change significantly to significant

6. PLOS authors have the option to publish the peer review history of their article (what does this mean?). If published, this will include your full peer review and any attached files.

Reviewer #1: No

Reviewer #2: No

---

## [Author Response · Author response to Decision Letter 0]

29 May 2022

We would like to thank the editor and reviewers for their suggestions. 

Please find our responses below. 

To the editor: 

1. New manuscript file format was changed to meet the PLOS ONE style. 

2. We updated the competing of interests as suggested by the editor: 

Competing interests: 

YZ, KZ, and XM are employees of GenomCan Inc.. DZ is an employee of Chengdu CapitalBio Technology Co., Ltd.. The remaining authors do not have any conflicts of interest to disclose. This does not alter our adherence to PLOS ONE policies on sharing data and materials. 

3. Captions for Supporting Information files were added at the end of manuscript. 

We added 2 supplementary figures and captions are provided at the end.

“S1 Fig. The distribution of fetal fraction grouped by the gestational hypertension status. Density plot of fetal fraction from gestational hypertension samples and normal samples in 2 different colors. Dotted lines represent the mean values of fetal fraction from the 2 groups.

S2 Fig. The distribution of fetal fraction grouped by the oligohydramnios status. Density plot of fetal fraction from oligohydramnios samples and normal samples in 2 different colors. Dotted lines represent the mean values of fetal fraction from the 2 groups.”

Reviewer 1:

1. Are there any indications for performing NIPT in this study? Do you provide genetic counseling for those who wish to do so?

Apologies that we didn't make the inclusion criteria very clear. 

This study is based on a screening test following the inclusion guideline by National Health and Family Planning Commission of the People's Republic of China. [Technical specifications for noninvasive prenatal screening and diagnosis using cell-free fetal DNA. (in Chinese) http://www.nhc.gov.cn/ewebeditor/uploadfile/2016/11/20161111103703265.docx ]. No indications required for the test, but conditions listed by the guideline were excluded.

In this study, NIPT positive samples were excluded, and patients with history of chronic hypertension or diabetes were excluded. And genetic counseling was provided before taking the NIPT test and when returning positive NIPT results. We have changed the description in materials and methods section accordingly. 

" We performed a single-institution retrospective cohort study of all women that underwent NIPT screening from May 2018 to May 2020 at Mianyang People’s Hospital. Women were recruited without any prior indication of genetic diseases or fetal structural abnormalities, following the guideline by the National Health and Family Planning Commission of China (http://www.nhc.gov.cn/ewebeditor/uploadfile/2016/11/20161111103703265.docx). Genetic counseling was provided before the NIPT test and when returning the NIPT result. Only women who had singleton pregnancies and showed no chromosomal aneuploidies from NIPT and with valid fetal fraction estimation and pregnancy complications information were included. Further exclusion criteria were: (1) women with multiple pregnancies; (2) women with missing information on whether singleton or multiple pregnancy; (3) women with missing fetal fraction information; (4) women with a history of chronic hypertension or diabetes."

2. Why did you perform NIPT around 17 weeks of gestation instead of 10-14 weeks?

As per the China Government guideline listed above, NIPT should be taken from 12+0-22+6 weeks. And most patients came for genetic consulting during first pregnancy visit around 12 weeks, then came for the NIPT test during next hospital visit, which is usually 5-6 weeks later, which lead to a mean of ~17 weeks. 

3. Please indicate in the Materials and methods section at which gestational week the diagnosis of oligohydramnios was made.

We have added the gestational week in the methods section. 

"Oligohydramnios was defined a single deepest vertical pocket (DVP) less than 2cm on ultrasound examination at 28-40 weeks."

4. Could you list the total number of NIPT performed, the positive rate, and the unreportable results rate in the figure?

We fixed Fig 1 to reflect the whole filtering process and the numbers. 

“There are 3834 women who underwent NIPT at Mianyang People’s Hospital between May 2018 and May 2020. Twenty-six women had NIPT chromosomal abnormality (a positive rate of 0.68%), which were excluded from the study. We also excluded 8 women with a history of chronic hypertension or diabetes. We finally obtained 3534 cases in the study after further excluding 266 patients without NIPT results, with multiple pregnancy and with missing information on whether singleton or multiple pregnancy. Details regarding the excluded cases were shown in Fig 1.”

5. It is speculated that the difference in the pathogenesis of HDP and preeclampsia affects the results of FF. What would you think about placental remodeling in early gestational age?

Thanks for the suggestion. We added the following text in the discussion. 

“We speculate that the pathophysiologic differences in women with preeclampsia and those with gestational hypertension could lead to different association results. Early placental arteriolopathy are more prevalent in women with preeclampsia[29]. The apoptosis process involved could potentially raise the fetal fraction of cfDNA[30].”

6. How do you think the FF tends to be lower in pregnant women with oligohydramnios?

We added the following discussion. 

“Although the exact cause of oligohydramnios is complicated and still unclear, we speculated that subtle placental structural abnormalities could potentially cause oligohydramnios. This could also potentially reduce the release of fetal cfDNA into mother's blood, resulting in low fetal fraction of cfDNA.”

7. Do you think there is a background of racial differences or genetic/environmental influences that cause preeclampsia?

Thanks for pointing this out. We included a bit discussion on this topic in the updated manuscript. 

 “There are findings indicating the racial differences in the incidence rate of preeclampsia[31,32], but how this links to the genetic or environmental causes is still an open question. Finding the exact cause of preeclampsia is still a big scientific challenge. Further studies are merited to find the mechanisms behind the associations of fetal fraction with the different categories of HDP.”

Reviewer 2:

1. The significance and odds ratio improved dramatically when fetal fraction was dichotomized using the 25th percentile, but no explanation is provided for why this cutoff was chosen.

We explicitly explained our choice in 2 places in the revision. 

Lines 179-184:

“The dichotomization cut-off point for fetal fraction was at the 25th percentile, namely 15.15%. The choice of the 25th percentile cut-off point is based on the results from the distributions of fetal fraction stratified by the traits showing significant associations with continuous fetal fraction, which is mentioned below, as well as the existing literature [11]. The 25th percentile cut-off is also widely used in stratifications in statistical analysis.”

Lines 208-211:

“The fetal fraction distributions comparing women with gestational hypertension versus normal women, women with oligohydramnios versus normal women (S1 and S2 Figs.) reveal that the maximum segregation of distribution occurs around the 25th percentile (fetal fraction = 15.15%).”

2. The paper may benefit from more figures showing a comparison of the fetal fraction distributions in the gestation hypertension and normal populations.

“The fetal fraction distributions comparing women with gestational hypertension versus normal women, women with oligohydramnios versus normal women (S1 and S2 Figs.) revealed that the maximum segregation of distribution occurs around the 25th percentile (fetal fraction = 15.15%).”

3. The fetal fraction calculation was described very loosely. Either more detail should be provided or a reference to a paper than employed a similar method should be provided.

“The fetal fraction was estimated based on the different fragment length distributions between maternal and fetal cfDNA. Fetal cfDNA fragment tend to have a higher proportion of short plasma DNA fragments (∼130–140 bp; region A) and a lower proportion of long plasma DNA fragments (∼155–175 bp; region B). Locally weighted scatterplot smoothing (LOESS) regression was applied to fit the fetal fraction against reads ratio in features A and B. Male fetuses with fetal fraction estimated from Y chromosome reads were used to train the model parameters[17].”

4. How the control group was defined for the statistical analysis. When evaluating the relationship between gestational hypertension and fetal fraction, were all other diagnoses included? Or only the gestational hypertension samples and the normal samples?

We added a clarification in the method section:

“For all association tests with each complication, only normal samples were included in the control group.”

5. The data was provided, but some more detail on how to analyze it to recreate the data in the paper would be useful. For example, it was not clear how to reproduce the data filtering from the flow chart provided. A list of issues is provided below.

• Figure 1 states there are 3808 total individuals, but the provided data set has 3800 rows.

• Figure 1 states that 226 samples were excluded due to Not singleton pregnancy, however the data has 228 samples with twin == 2.

• Patients with history of hypertension (7) and diabetes (1) are not indicated. Perhaps this explains the difference between the 3808 and 3800?

We made the filtering steps more specific in the revised Fig 1. 

We excluded 8 patients with a history of chronic diseases (7 with hypertension and 1 with diabetes) in the dataset provided. 

We firstly excluded patients whose NIPT results were not available. Then, we excluded patients without singleton pregnancies. There were two patients with twin pregnancies as well as missing NIPT results. Thus, it shows the 2-case difference in calculation of 226 vs 228. 

We fixed all grammar errors in the manuscript as suggested by the reviewer and other errors we identified. 

Line 52: “leading causes” should be “leading cause.” Also, the statement is fairly obvious. “Have been” usually implies some passage of time. Consider: “Pregnancy related complications are the leading cause”. 

“Pregnancy-related complications are the leading cause of the morbidity and mortality of pregnancy.”

Lines 53-54: Change “due to the complications” to “due to complications” 

“Nearly one fifth of all pregnant women are hospitalized before delivery due to complications [1]. ”

Lines 54-56: Consider rephrasing. Start the sentence with: “Common pregnancy related complications include...” 

“Common pregnancy-related complications include hypertensive disorders of pregnancy (HDP) and gestational diabetes [2–4].”

Line 61: cfDNA is a candidate biomarker for pregnancy related complications. 

“Fetal cell-free DNA (cfDNA) is a candidate biomarker for pregnancy related complications.”

Lines 61-62: Cell free DNA is composed of DNA fragments released into the blood plasma. 

“Cell free DNA in the blood plasma of pregnant women is composed of DNA fragments mainly from the mother herself, but also from the fetus.”

Lines 64-66 : Fetal cfDNA is released into the maternal circulation via apoptosis of placental villous trophoblasts throughout gestation. 

“Fetal cfDNA is released into the maternal circulation via apoptosis of placental villous trophoblasts throughout gestation [7].”

Line 69: Remove we: “... NIPT, fetal and maternal cfDNA can be sequenced...”

“With the development and commercial availability of NIPT, fetal and maternal cfDNA can be sequenced efficiently and cost-effectively through massively parallel DNA sequencing (MPS).”

Lines 76: Replace “the links of fetal fraction with” with “the relationship between fetal fraction and” Line 78: Consider replacing conflicting with inconsistent. 

“Previous studies have investigated the relationship between fetal fraction and various pregnancy complications including HDP [9–13], gestational diabetes [14], and preterm birth [10–12,15].”

Line 78: Consider replacing conflicting with inconsistent.

“However, the significance and directionality of these associations were inconsistent.”

Line 90: Replace countries with populations or perhaps cohorts. 

“However, the significance and directionality of these associations were inconsistent. In addition, most of previous studies were conducted in European and American populations.”

Line 80: Replace “were limited” with “are limited”

“Studies in East Asians, especially in Chinese populations are limited.”

Line 88: Replace undergoing with that underwent Line 89 : Replace receiving with received 

“We performed a single-institution retrospective cohort study of all women that underwent NIPT screening from May 2018 to May 2020 at Mianyang People’s Hospital. Women were recruited without any prior indication of genetic diseases or fetal structural abnormalities, following the guideline by the National Health and Family Planning Commission of China (http://www.nhc.gov.cn/ewebeditor/uploadfile/2016/11/20161111103703265.docx). Genetic counseling was provided before the NIPT test and when returning the NIPT result. Only women who had singleton pregnancies and showed no chromosomal aneuploidies from NIPT and with valid fetal fraction estimation and pregnancy complications information were included. Further exclusion criteria were:”

Line 99: add “the” before instrument and reagents 

Line 101: MPS already defined as an acronym on line 70 

“In brief, the BioelectronSeq 4000 sequencing instrument and the Sequencing Reaction Universal Kit (CapitalBio, Dongguan, People's Republic of China) were used for MPS of plasma cfDNA fragments.”

Lines 102-105: The method for estimating fetal fraction is not adequately described. Simply writing “estimated using different length distributions” is too vague. Either provide more detail, or provide a reference to a previous method. The reads ratio has not been defined. 

“The fetal fraction was estimated based on the different fragment length distributions between maternal and fetal cfDNA. Fetal cfDNA fragment tend to have a higher proportion of short plasma DNA fragments (∼130–140 bp; region A) and a lower proportion of long plasma DNA fragments (∼155–175 bp; region B). Locally weighted scatterplot smoothing (LOESS) regression was applied to fit the fetal fraction against reads ratio in features A and B. Male fetuses with fetal fraction estimated from Y chromosome reads were used to train the model parameters[17].”

Line 103: Add fragment before length 

“The fetal fraction was estimated based on the different fragment length distributions between maternal and fetal cfDNA.”

Line 104: Male fetus should be plural 

“Male fetuses with fetal fraction estimated from Y chromosome reads were used to train the model parameters[17].”

Line 114: Replace “Our” with “The” 

“The outcomes of interest were pregnancy-related complications,”

Line 126: who did not have.. 

“Gestational diabetes was defined as diabetes firstly seen in a pregnant woman who had no diabetes before the pregnancy”

Line 132: Replace “our” with “the” 

“We analyzed the variable of interest, fetal fraction, as both continuous and dichotomous variables.”

Lines 136-137: Please elaborate on how the dichotomous threshold was determined from the FF distribution in the cohort. 

We explicitly explained our choice in 2 places in the revision. 

Lines 179-184:

“The dichotomization cut-off point for fetal fraction was at the 25th percentile, namely 15.15%. The choice of the 25th percentile cut-off point is based on the results from the distributions of fetal fraction stratified by the traits showing significant associations with continuous fetal fraction, which is mentioned below, as well as the existing literature [11]. The 25th percentile cut-off is also widely used in stratifications in statistical analysis.”

Lines 208-211:

“The fetal fraction distributions comparing women with gestational hypertension versus normal women, women with oligohydramnios versus normal women (S1 and S2 Figs.) reveal that the maximum segregation of distribution occurs around the 25th percentile (fetal fraction = 15.15%).”

Line 148: Unclear what “fetal fraction as a whole” means. If you are not referring to fetal fraction, but rather the combined set of gestational hypertension and preeclampsia, then please clarify. 

Yes. It means HDP. 

“We firstly assessed the association between fetal fraction with HDP, combining gestational hypertension and preeclampsia.”

Lines 148: Replace “Then we went further and” with “We also”

“We also evaluated the relationships of fetal fraction with gestational hypertension and preeclampsia separately,”

Line 153: replace “our” with “this” and remove “also”

Line 155: Replace “proper” with “optimal.” 

“To examine this hypothesis, we performed stratified analyses based on gestational age at NIPT: first trimester (10-14.9 weeks), second trimester (15 and 22 weeks) and third trimester (>22 weeks) in order to find the optimal time to perform the NIPT screening tests.”

Lines 151-155: You discuss stratifying by trimester and identifying the best trimester for testing. Was this done separately for each pregnancy complication? This sentence comes directly after a discussion of hypertensive disease, so it’s unclear if it is just for that complication or for all. If it is for all, consider starting a new paragraph. 

It is for all traits. We started as a new paragraph.

Line 162. Generally, numbers greater than 1,000 are written as numbers 

Line 162: Replace “our institution” with the institution name. 

“There are 3834 women who underwent NIPT at Mianyang People’s Hospital between May 2018 and May 2020. Twenty-six women had NIPT chromosomal abnormality (a positive rate of 0.68%), which were excluded from the study. We also excluded 8 women with a history of chronic hypertension or diabetes. We finally obtained 3534 cases in the study after further excluding 266 patients without NIPT results, with multiple pregnancy and with missing information on whether singleton or multiple pregnancy.”

Line 170: Replace “For our population, the fetal fraction ranged from” with “The fetal fraction within the study population ranged from” 

Line 171: Standard deviation needs % units. 

“Fetal fractions within the study population ranged from 4.32% to 50.38%, with a mean value of 18.48% and a standard deviation of 5.28%.”

Lines 172-175: This sentence is very hard to follow. It has too many commas and it isn’t clear what the subject of the sentence is. If the sentence is meant to describe how the FF cutoff was established, as verb is missing. For example: “The identification of the cut-off point was based” 

“The choice of the cut-off point is based on the results from the distributions of fetal fraction stratified by the traits showing significant associations with continuous fetal fraction, which is mentioned below, as well as previous literatures [11]. The 25th percentile cut-off is also widely used in stratifications in statistical analysis.”

Lines 176-178: The relationship between FF, gestational age, and BMI is well established. It may be worth mentioning this. 

“We found statistically significant differences between the two groups regarding maternal BMI, gestational age at delivery, and method of delivery. This is not surprising given that associations of fetal fraction with gestational age and maternal BMI are well established in previous publications [18,19].”

Line 195: add “a”: “For a one percent…”

“For every one percent increase of fetal fraction,”

Line 272: Change significantly to significant 

“In this retrospective cohort study, we observed significant inverse associations of fetal fraction with the risk of hypertensive disorders of pregnancy (HDP) and oligohydramnios after adjusting for potential confounders.”

---

## [Decision Letter · Decision Letter 1]

27 Jun 2022

The association between fetal fraction and pregnancy-related complications among Chinese population

PONE-D-21-33442R1

Dear Dr. Yin,

We’re pleased to inform you that your manuscript has been judged scientifically suitable for publication and will be formally accepted for publication once it meets all outstanding technical requirements.

Kind regards,

Badri Padhukasahasram

Academic Editor

PLOS ONE

Additional Editor Comments (optional):

Reviewers' comments:

Reviewer's Responses to Questions

**Comments to the Author**

1. If the authors have adequately addressed your comments raised in a previous round of review and you feel that this manuscript is now acceptable for publication, you may indicate that here to bypass the “Comments to the Author” section, enter your conflict of interest statement in the “Confidential to Editor” section, and submit your "Accept" recommendation.

Reviewer #2: All comments have been addressed

2. Is the manuscript technically sound, and do the data support the conclusions?

Reviewer #2: Yes

3. Has the statistical analysis been performed appropriately and rigorously? 

Reviewer #2: Yes

4. Have the authors made all data underlying the findings in their manuscript fully available?

Reviewer #2: Yes

5. Is the manuscript presented in an intelligible fashion and written in standard English?

Reviewer #2: Yes

6. Review Comments to the Author

Reviewer #2: The authors have responded to my comments in their entirety. I have no additional comments to make.

7. PLOS authors have the option to publish the peer review history of their article (what does this mean?). If published, this will include your full peer review and any attached files.

Reviewer #2: No

---

## [Editor Report · Acceptance letter]

4 Jul 2022

PONE-D-21-33442R1 

The association between fetal fraction and pregnancy-related complications among Chinese population 

Dear Dr. Yin:

I'm pleased to inform you that your manuscript has been deemed suitable for publication in PLOS ONE. Congratulations! Your manuscript is now with our production department. 

Kind regards, 

on behalf of

Dr. Badri Padhukasahasram 

Academic Editor

PLOS ONE